# Determination of Methyl Methanesulfonate and Ethyl Methylsulfonate in New Drug for the Treatment of Fatty Liver Using Derivatization Followed by High-Performance Liquid Chromatography with Ultraviolet Detection

**DOI:** 10.3390/molecules27061950

**Published:** 2022-03-17

**Authors:** Yue Wang, Jing Feng, Song Wu, Huihui Shao, Wenxuan Zhang, Kun Zhang, Hanyilan Zhang, Qingyun Yang

**Affiliations:** State Key Laboratory of Bioactive Substance and Function of Natural Medicines, Institute of Materia Medica, Chinese Academy of Medical Sciences & Peking Union Medical College, Beijing 100050, China; wangyue@imm.ac.cn (Y.W.); fengjinga@imm.ac.cn (J.F.); ws@imm.ac.cn (S.W.); shaohuihui@imm.ac.cn (H.S.); wxzhang@imm.ac.cn (W.Z.); aimokun@163.com (K.Z.); zhanghanyilan@imm.ac.cn (H.Z.)

**Keywords:** genotoxic impurities, derivatization, HPLC-UV, sodium dibenzyldithiocarbamate

## Abstract

A new derivatization high-performance liquid chromatography method with ultraviolet detection was developed and validated for the quantitative analysis of methanesulfonate genotoxic impurities in an innovative drug for the treatment of non-alcoholic fatty liver disease. In this study, sodium dibenzyldithiocarbamate was used as a derivatization reagent for the first time to enhance the sensitivity of the analysis, and NaOH aqueous solution was chosen as a pH regulator to avoid the interference of the drug matrix. Several key experimental parameters of the derivatization reaction were investigated and optimized. In addition, specificity, linearity, precision, stability, and accuracy were validated. The determined results of the samples were consistent with those obtained from the derivatization gas chromatography–mass spectrometry analysis. Thus, the proposed method is a reliable and practical protocol for the determination of trace methanesulfonate genotoxic impurities in drugs containing mesylate groups.

## 1. Introduction

In the synthetic process of active pharmaceutical ingredients (APIs) containing mesylate groups, methyl methanesulfonate (MMS) and ethyl methanesulfonate (EMS) inevitably reside in the final products as potential genotoxic impurities (PGIs) [1,2]. These PGIs could directly alkylate with biological macromolecules, leading to gene mutation and tumorigenesis, even at trace levels [3]. They also play a genotoxic role in bacteria and mammalian cells [4,5,6]. This finding is related to the withdrawal of Viracept of the Roche company from European markets due to excessive EMS residue [7], which is generated by the reaction of ethanol resided by equipment cleaning and methanesulfonic acid (MSA). Therefore, monitoring and controlling these PGIs at appropriate and safe levels is highly important in the development and manufacturing of APIs for regulatory requirements. The European Medicines Agency (EMEA) [8], the Food and Drug Administration (FDA) [9], and the International Council for Harmonisation of Technical Requirements for Pharmaceuticals for Human Use (ICH) M7 (R1) [10] issued relevant regulations on the limits of PGIs. For some genotoxic impurities without specific toxicological data, the limits could be controlled by the threshold of toxicological concern (TTC), which is 1.5 μg·day^−1^ for long-term treatment, with higher limits for short-term clinical treatment.

IMH, 2,4,4′-trimethoxy-5,6,5′,6′-bis(methylenedioxy)-2′-morpholine methylbiphenyl methanesulfonate, shown in Figure 1, is an innovative chemical drug for the treatment of non-alcoholic fatty liver disease (NAFLD), and it has almost completed preclinical studies and will soon be submitted for clinical trials. It is a kind of mesylate drug, and MSA was used as a counter ion to form salt during the synthesis of APIs. MSA is commonly used as an acid for the salt formation of alkaline drugs or employed as a reagent in synthesis to improve the solubility and stability of drugs, thus improving druggability [11]. However, MSA can easily react with short-chain alcoholic agents to form potential genotoxic methanesulfonates [12]. Given that methanol and ethanol are frequently used as solvents for crystallization or purification in the process of API synthesis, MMS and EMS are two common impurities. Therefore, a sensitive and validated method must be developed for the reliable estimation of MMS and EMS in IMH to ensure the safety of drug administration. The relevant guidelines of PGIs proposed a TTC of 1.5 μg·day^−1^ for drug formulations. Taking the maximum daily dosage (100 mg) into account, the estimated permitted level of these impurities in the IMH API is 15 ppm.

For the determination of MMS and EMS, the common analytical methods include gas chromatography (GC) [13,14,15] and high-performance liquid chromatography (HPLC) [16,17,18]. For ppm-level detection, direct injection GC methods suffer from severe contamination issues because of the high concentrations of APIs onto a GC column. Thus, the headspace injection method is often used. However, MMS and EMS do not evaporate easily due to their boiling points of more than 200 °C. The European Pharmacopoeia (EP) [19] recommends NaI as a derivatization agent to generate volatile iodoalkane for injection. In addition, thiocyanate [20] and pentafluorothiophenol [21] can be used as derivatization reagents to form volatile substances with MMS and EMS. LC columns can usually tolerate a much higher sample loading than GC capillary columns. However, determining MMS and EMS by the HPLC–UV method directly is not feasible due to the lack of UV chromophore. Therefore, adding UV chromophore by pre-column derivatization is required to improve the feasibility and sensitivity of the HPLC–UV assay. Zhou et al. [22] successfully determined the content of MMS and EMS in MSA by the HPLC–UV method with the use of sodium *N*,*N*-diethyldithiocarbamate (DDTC) as a derivatization reagent. However, this method could not be used for the determination of PGIs in API because of the interference of the drug matrix. Subsequently, other research [23] used headspace single-drop microextraction technology to determine alkyl methanesulfonate in API by HPLC–UV. However, this method was complicated, and it required a two-step derivatization reaction.

Herein, a new derivatization HPLC–UV method was established for the analysis of MMS and EMS in IMH API using sodium dibenzyldithiocarbamate (BDC) as the derivatization agent. NaOH aqueous solution was chosen as the pH regulator, and it effectively avoided the interference of the drug matrix. The optimal reaction conditions were determined by selecting different reaction solvents and reaction environments. The methodology validation of the whole method proved that this proposed method was specific, accurate, rapid, and sensitive for the determination of MMS and EMS in API. Several batches of APIs were chosen to verify the reliability of the proposed method. The results were further compared with those obtained by the derivatization GC–MS method mentioned in EP 10.0 [19]. The results indicated that the new derivatization HPLC–UV method is a reliable method for the detection of trace MMS and EMS in IMH API. Furthermore, this method has a high reference value for the quality control of other drugs containing MMS and EMS impurities.

## 2. Results and Discussion

### 2.1. Optimization of Derivatization Reactions

#### 2.1.1. Selection of Derivatization Reagent

This work aimed to develop sensitive and reliable LC–UV methods for the determination of PGIs in IMH API. For this purpose, several derivatization reagents were tested and compared, including sodium *N*,*N*-diethyldithiocarbamate (DDTC), sodium thiophenolate (TPO), and sodium dibenzyldithiocarbamate (BDC). The structures of the derivatization reagents are shown in Figure 1. The UV maximum absorption wavelength and limit of quantitation (LOQ) of the MMS and EMS derivatives obtained by reaction with the above three derivatization reagents are listed in Table 1. The UV wavelength of the BDC derivatives was 280 nm, longer than that of the other two derivatives. This finding indicated flatter baselines and better line shape on the chromatogram. Furthermore, the LOQ of the MMS and EMS derivatives with BDC were lower than those of the other derivatives. Meanwhile, the interference of the EMS derivatives existed in TPO, and the retention time of the BDC derivatives was shorter than that of the DDTC derivatives. Therefore, BDC was selected as the derivatization reagent for further optimization. The scheme of the derivatization reaction is shown in Figure 2.

#### 2.1.2. Selection of Derivatization Solvent

Nucleophilic substitution reactions often occur in aprotic solvents. In this study, several aprotic solvents, including acetonitrile (ACN), *N*,*N*-dimethylformamide (DMF), *N*,*N*-dimethylacetamide (DMA), and dimethyl sulfoxide (DMSO), were screened for the experiment as part of the preliminary work. The derivatization reagents were dissolved in four solvents at the same concentration for determination. As the peak areas in the four solvents have a slight difference, the solvent with the better peak type was selected by peak height. The HPLC chromatograms of different derivatization solvents are shown in Appendix A. As shown in Figure 3, the peak heights of the MMS and EMS derivatives were higher in ACN, indicating that the yields of the derivatization reactions were affected by the solvents to some extent. Thus, ACN was chosen as the derivatization solvent.

#### 2.1.3. Optimization of Derivatization Reaction Conditions

The effects of different reaction conditions are shown in Figure 4. The results showed that heating had a higher reactivity than no treatment, light, and ultrasonic, especially for the derivatization reaction of EMS. Then, the reaction temperatures, reaction times, and concentrations of the derivatization reagent solutions were optimized further under heating conditions (as shown in Appendix A). As shown in Figure 5A,B, for the derivatization reaction of EMS, the peak areas of the derivatives reached a maximum when the reaction condition was 80 °C. Meanwhile, the reaction temperature, reaction time, and concentration of the derivatization solution had a slight influence on the derivatization reaction of MMS. With such conditions, the responses of the derivatives were enhanced by increasing the concentration of the derivatization reagent from 0.5 mg·mL^−1^ to 3.0 mg·mL^−1^ (Figure 5C), and then no significant change could be further observed. As a consequence, the optimal derivatization conditions were determined as follows: BDC (3 mg·mL^−1^) in ACN for 2 h at 80 °C.

#### 2.1.4. Optimization of Other Derivatization Parameters

According to the above optimized conditions, the recoveries of the derivatives were very low because of the API matrix interference. Some methods, such as *n*-hexane extraction [24], solid-phase extraction (SPE) [25], and matrix precipitation [26], are used to remove the sample matrix. In the present study, various methods were adopted to remove the sample matrix. Unfortunately, all the test results were unsatisfactory.

The acidity of API was considered to change the pH condition of the reaction solution, which was unfavorable to the reaction. Thus, several alkaline reagents were adopted to adjust the pH condition of the reaction solution, including the inorganic strong base, NaOH; the strong base weak acid salt, Na_2_CO_3_; and the organic base, triethylamine (Appendix A). The effects of the different alkaline reagents (Figure 6D) and concentrations (Figure 6A–C) on the reaction were investigated and compared.

With the increase in alkali concentration, the peak area of the derivative increased gradually with the increase in solution pH. When the pH reached 6.0–7.0, the peak area of the derivative tended to be stable. As triethylamine can react with MMS, promoting the desired derivatization reaction was difficult, and the solid will precipitate after adding Na_2_CO_3_. The tolerance of the chromatographic column and the solubility of API should be considered. Therefore, 0.5 mL of the 40 mg·mL^−1^ NaOH solution was the suitable additive.

### 2.2. Method Validation and Application

#### 2.2.1. HPLC Development

For this study, C8 and C18 stationary phases with different carbon loadings were adopted for the method development in the initial stage of the experiment. Given that MSA easily reacts with short-chain alcoholic agents to form methanesulfonates, the ACN–water mobile phase system was selected instead of the methanol–water system to avoid false-positive results. Different proportions of the ACN–ammonium acetate solution were tested. Finally, a good peak separation was observed on the SunFire C18 column (250 mm × 4.6 mm, 5 μm particle size) by using the ACN–5 mmol·L^−1^ ammonium acetate solution at a constant proportion of 20:80 (*V*/*V*) as the mobile phase. The method demonstrated good separation among the impurities with a short running time, and it could resist the interference of the API matrix effectively. In addition, the maximum absorption wavelengths of the MMS and EMS derivatives were 279.3 nm and 281.7 nm (Figure 7), respectively. Thus, 280 nm was selected as the determination wavelength.

#### 2.2.2. Method Validation

HPLC chromatograms (at 280 nm) of the blank solution, sample solution, standard solution, and spiked sample solution under optimal conditions are shown in Figure 8. The API and derivatization reagent peaks did not interfere with the peaks of the MMS and EMS derivatives.

The data from the validation experiments are summarized in Table 2. Linearity was evaluated by preparing mixed standard solutions containing MMS and EMS at different concentration levels. A linearity curve was plotted, and the slope, intercept, and correlation coefficient were obtained by a least-square linear regression analysis. The linearity was satisfactorily illustrated with a seven-point calibration graph. The LOQ values of the MMS and EMS derivatives were 0.15 ng·mL^−1^ and 0.30 ng·mL^−1^, equivalent to 0.3 ppm and 0.6 ppm, respectively. Precision was estimated by the sample solution added with known concentrations of the mixed standard MMS and EMS. The RSD values for the six repeated injections were 3.23% and 1.66%. The RSD values of the 12 solutions of the two instruments were 3.50% and 2.39%, indicating that the intermediate precision was good. The stability of the same spiked sample solution after the derivatization was observed at different time points within 24 h at room temperature, and the RSD value was within 3%. In addition, the accuracy was determined through spiked recovery experiments, and the average recovery rates of the four spiked concentrations levels in triplicate were calculated. Good recoveries in the range of 99%–101% with RSD values below 5% were achieved.

#### 2.2.3. Sample Analysis

The validated derivatization HPLC–UV method was applied to measure the methanesulfonate PGIs in the three batches of IMH API samples and compared with classical derivatization GC–MS analysis (as shown in Appendix A). The results are listed in Table 3. The levels of MMS were below the defined acceptable TTC limits, and EMS was not detected in all the batches of API samples, indicating that all the impurities are well controlled.

## 3. Materials and Methods

### 3.1. Materials, Chemicals, and Reagents

The bulk of the API of IMH (batch nos. 20180608, 20180918, and 20181026) was produced in the authors’ laboratory. MMS (99%) and EMS (99%) were purchased from Alfa Aesar (Shanghai, China). DDTC (99%) and BDC (98%) were obtained from Aladdin (Shanghai, China). HPLC-grade *N*,*N*-dimethylacetamide (DMA), dimethyl sulfoxide (DMSO), and *n*-hexane (95%) were purchased from Innochem (Beijing, China). HPLC-grade acetonitrile (ACN) was acquired from Fisher (Shanghai, China). Sodium thiophenolate (TPO, 97%) and NaOH were also purchased from Innochem (Beijing, China). Ammonium acetate (98%) was provided by SINOPHARM (Beijing, China). *N*,*N*-dimethylformamide (DMF) was supplied from Tong Guang (Beijing, China). Dichloromethane (99.5%) was obtained from Xilong Scientific (Guangdong, China). Triethylamine (99.7%) was purchased from J&K Scientific (Beijing, China), and sodium carbonate (99.8%) was purchased from Beijing Chemical Works (Beijing, China). The purified water was purchased from Wahaha (Hangzhou, China).

### 3.2. Instrumentation and Chromatographic Conditions

HPLC analysis was performed using Waters e2695 equipped with a Waters 2998 Photodiode Array Detector (Waters, Milford, MA, USA). Chromatographic separations were achieved using a SunFire C18 column (250 mm × 4.6 mm, 5 μm particle size) maintained at 30 °C. The mobile phase was a mixture of 5 mM ammonium acetate (mobile phase A) and ACN (mobile phase B) in a constant proportion of 20:80 (*V*/*V*) at a flow rate of 1.0 mL·min^−1^. The injection volume was set at 20 μL, and the detection wavelength was 280 nm. In the intermediate precision experiment, the liquid chromatograph was replaced by Thermo Ultimate 3000 (Thermo, Waltham, MA, USA).

GC–MS analysis [19] was performed using Thermo Scientific TRACE 1310/ISQ equipped with an electron ionization ion source (Thermo, Waltham, MA, USA). The ionizing energy was 70 eV. The compounds were separated on a polar-deactivated polyethyleneglycol column (30 m × 0.25 mm × 1 μm film). A 2 μL volume with a 1:20 split inlet was selected for injection. The static headspace conditions were as follows: an equilibration temperature of 60 °C, an equilibration time of 30 min, and a transfer-line temperature of 120 °C. The gas chromatographic conditions were an initial oven temperature of 40 °C (1 min) programmed to 130 °C at 10 °C·min^−1^. Helium was used as a carrier gas (flow rate of 0.5 mL·min^−1^). The injection port, ion source, and analyzer temperatures were 220 °C, 250 °C, and 200 °C, respectively.

### 3.3. Sample Preparation of Derivatization HPLC–UV Method

#### 3.3.1. Standard and Test Solutions

Stock solutions containing MMS and EMS were prepared at a concentration of 7.5 μg·mL^−1^ with ACN for method validation and analysis. The stock solutions of BDC were dissolved in ACN at a concentration of 3 mg·mL^−1^ as the derivatizing agent solution. NaOH solution in water was prepared at a concentration of 40 mg·mL^−1^.

#### 3.3.2. Derivatization Procedure

The optimal derivatization procedure was obtained by screening the derivatization solvent, conditions, and concentration of the derivatizing reagent. First, 0.5 mL of stock solution (7.5 μg of MMS and EMS per 1 mL), as described in Section 3.3.1, was added to a 5 mL volumetric flask, followed by 3 mL of the derivatizing reagent solution and 0.5 mL of water. The solution was diluted to a scale with ACN as a standard solution. An additional 3 mL of the derivatizing reagent solution was added to a 5 mL volume flask, followed by 0.5 mL of water; this solution was diluted to a scale with ACN as a blank control solution. The sample (250 mg) was weighed precisely and placed in a 5 mL volumetric flask; 3 mL of the derivatizing reagent solution and 0.5 mL of the NaOH solution (40 mg·mL^−1^) were added; finally, the mixture was diluted with ACN to a scale as a test solution. The contents of MMS and EMS in the test solution were calculated by the reference method. All the flasks were shaken well and then heated at 80 °C in a water bath for 2 h. After the reaction was completed, 20 μL was injected into the HPLC for determination. All the samples and standard solutions were filtered through 0.45 μm membrane filters before analysis.

#### 3.3.3. Method Validation

The determination method was validated in terms of specificity, linearity, precision, accuracy, and stability. A calibration plot was prepared by analyzing seven standard solutions containing MMS and EMS in the concentration ranges of 0.03 μg·mL^−1^–3.00 μg·mL^−1^ to establish linearity. The intercept, slope, and correlation coefficient were determined by linear regression. Precision was evaluated by IMH solutions added with known concentrations of the mixed standard MMS and EMS. Six solutions were prepared in parallel to obtain repeatability. Then, 12 samples were determined on two different instruments, and the intermediate precision was obtained. The results were estimated by calculating the relative standard deviation (RSD) values. The stability of the solution was evaluated by analyzing the peak area at 0, 2, 4, 8, 12, and 24 h, and then the RSD values were calculated. Finally, the accuracy of the method was determined via recoveries. In the recovery study, known amounts of MMS and EMS were added to the IMH solutions. The recoveries were calculated by comparing the experimental and theoretical values as follows: recovery (%) = 100 × (C − C_0_)/C_s_, where C is the total concentration after adding standards, C_0_ is the original concentration before addition, and C_s_ is the added concentration. The concentration of IMH was 50 mg·mL^−1^. Each of the above samples were repeated three times and treated with the established derivatization method. The appropriate amount of 30 ng·mL^−1^ solution and dilute was precisely measured with ACN stepwise, and LOQs were defined as the concentrations that could be detected and yield signal-to-noise (S/N) ratios of 10:1.

### 3.4. Sample Preparations of GC–MS Method

The solutions for the GC–MS method [19] were prepared as follows: approximately 25 mg of the IMH was accurately weighed, transferred into a 20 mL headspace vial, and then added with 0.5 mL of sodium iodide solution and 0.5 mL of the internal standard solution. The vial was sealed immediately with a polytetrafluoroethylene-coated silicon membrane and an aluminum cap. The conditions of static headspace were as follows: an equilibration temperature of 60 °C, an equilibration time of 30 min, and a transfer-line temperature of 120 °C.

## 4. Conclusions

In this study, a derivatization HPLC–UV method was successfully developed and validated for the quantitative analysis of MMS and EMS PGIs in an innovative methanesulfonate bulk drug for the treatment of NAFLD. BDC was used as the derivatization reagent for the first time to enhance the UV absorption of the MMS and EMS derivatives, and consequently to improve the sensitivity of the analysis. Furthermore, 10% NaOH aqueous solution was chosen as the pH regulator to avoid the interference of the drug matrix and improve the recovery of this method. Three batches of IMH API samples were chosen to verify the feasibility of the proposed method. Comparison of the derivatization HPLC–UV method with the derivatization GC–MS approach revealed that the two methods were almost identical. The proposed method could be used as a convincing supplement to the GC–MS method for the determination of methanesulfonate genotoxic impurities in pharmaceuticals. The new method could be applied to in-process monitoring of methanesulfonate PGIs during pharmaceutical manufacturing. As a versatile and convenient method, the proposed method has a high reference value for the quality control of other mesylate drugs. Therefore, this study could help ensure the safe use of these drugs during clinical treatments.

## Figures and Tables

**Figure 1 molecules-27-01950-f001:**
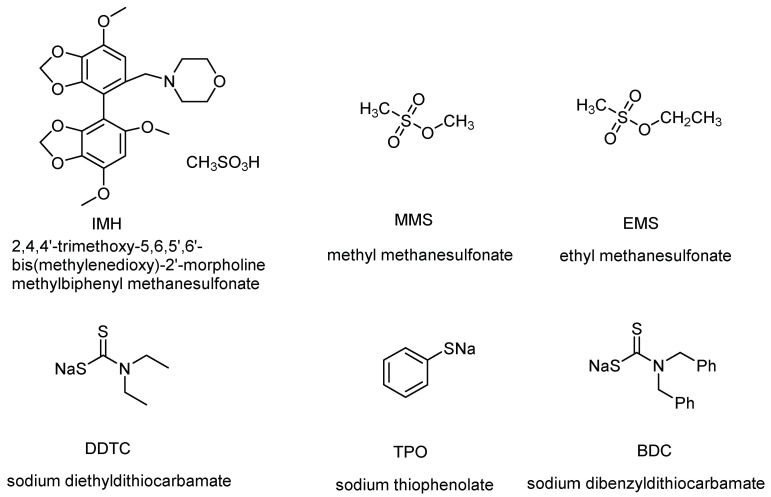
Chemical structures of IMH, MMS, EMS, and the derivatization reagents.

**Figure 2 molecules-27-01950-f002:**
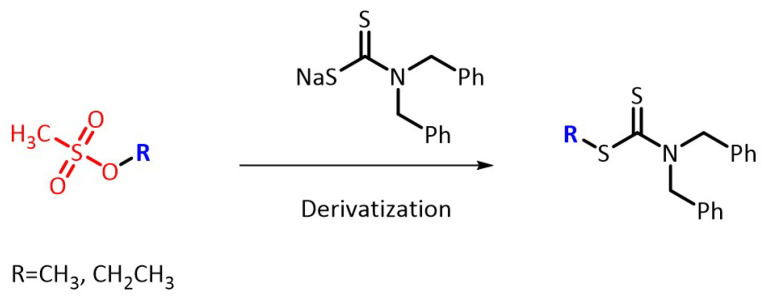
The derivatization reaction of the methanesulfonates with BDC.

**Figure 3 molecules-27-01950-f003:**
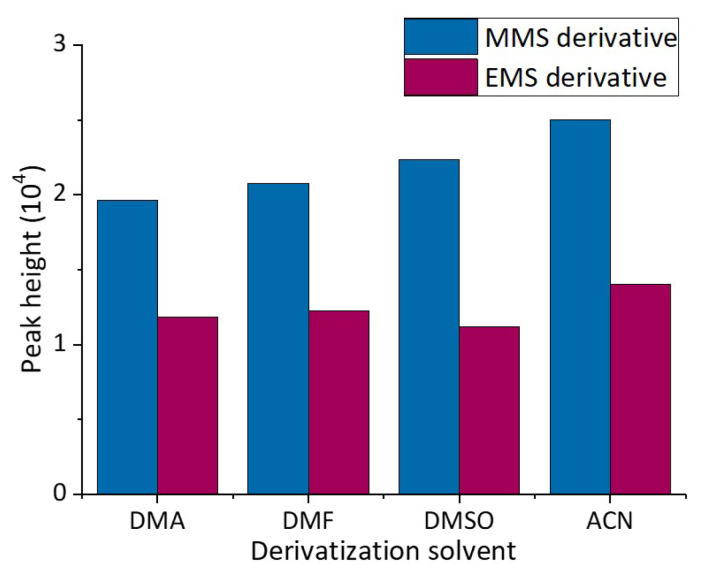
Different derivatization solvents.

**Figure 4 molecules-27-01950-f004:**
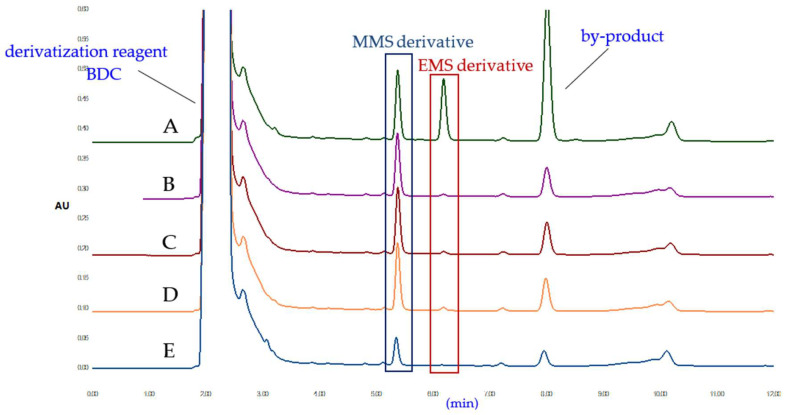
Chromatograms of the MMS and EMS derivatives in the different reaction conditions. (**A**) 80 °C 1 h; (**B**) ultrasonic 1 h; (**C**) light 1 h; (**D**) untreated 1 h; (**E**) untreated 0 h.

**Figure 5 molecules-27-01950-f005:**
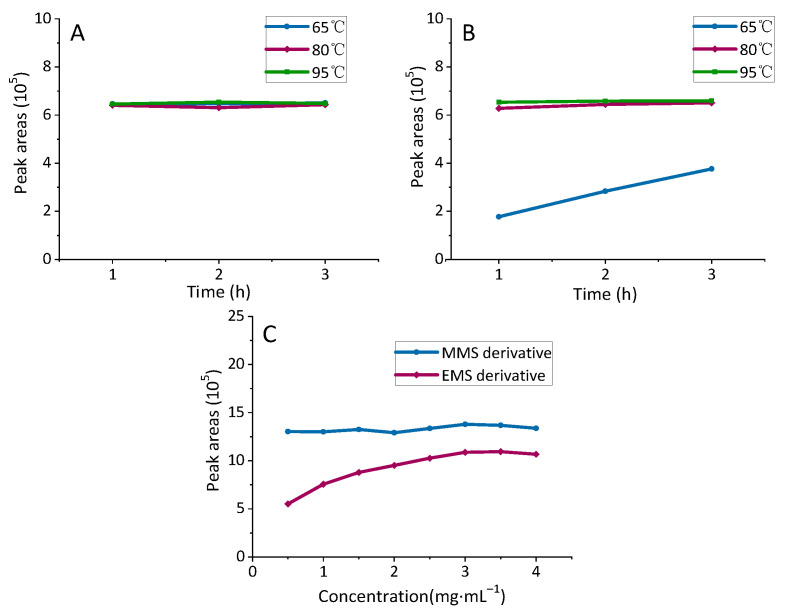
(**A**) Temperatures and times of the MMS derivatization reaction; (**B**) temperatures and times of the EMS derivatization reaction; (**C**) concentrations of the derivatization reagent.

**Figure 6 molecules-27-01950-f006:**
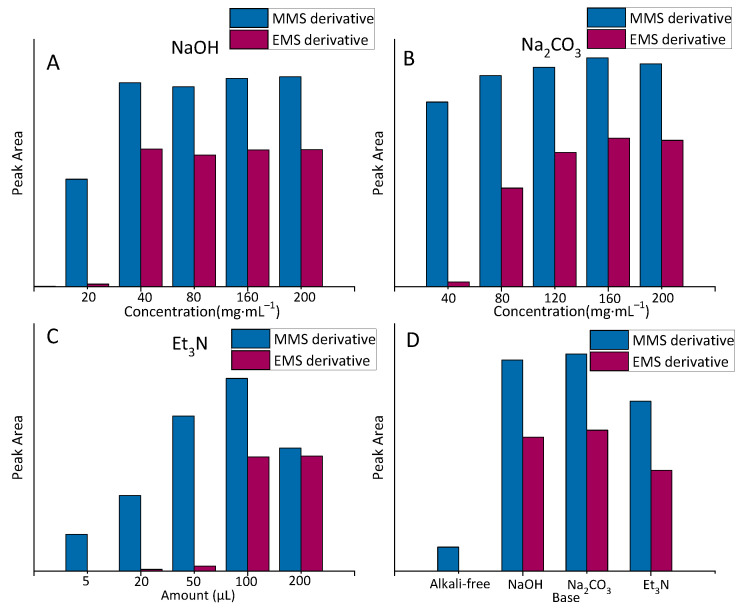
Types and concentrations of base. (**A**) Different concentrations of NaOH; (**B**) different concentrations of Na_2_CO_3_; (**C**) different concentrations of Et_3_N; (**D**) comparison of the results under the optimal concentration of the three bases.

**Figure 7 molecules-27-01950-f007:**
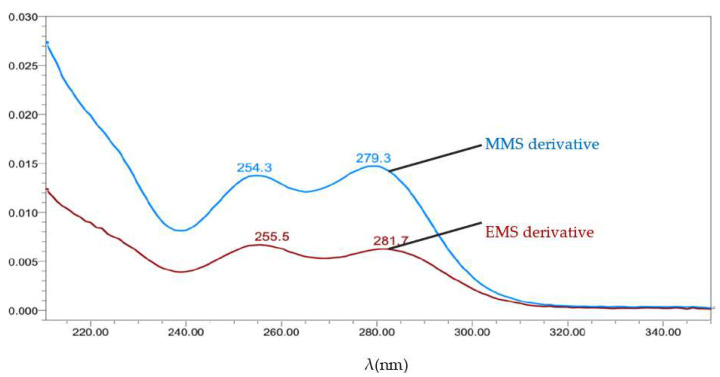
UV spectra of the MMS and EMS derivatives.

**Figure 8 molecules-27-01950-f008:**
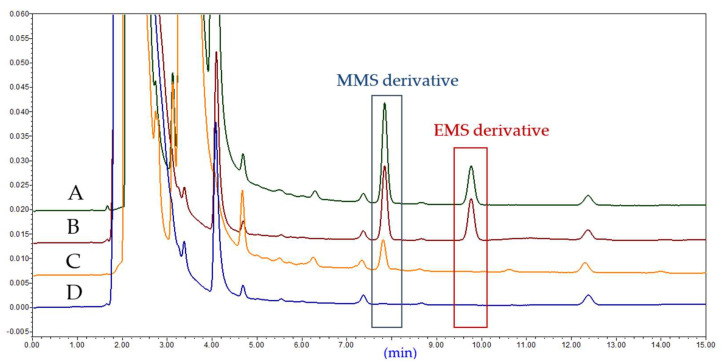
Typical chromatograms of the mixed 0.75 μg·mL^−^^1^ MMS and 0.75 μg·mL^−^^1^ EMS with BDC as the derivatization reagent. (**A**) Spiked sample; (**B**) impurity; (**C**) sample; (**D**) blank.

**Table 1 molecules-27-01950-t001:** A comparison of the derivatization reagents.

Derivatization Reagent	Sample	Feature	LOQ of Derivatives		Ref.
			MMS	EMS	
DDTC ^a^	Methanesulfonic acid	Simple, reliable, but not applicable to APIs due to matrix interference.	0.6 ppm	0.6 ppm	[22]
	Imatinib mesylate Levofloxacin mesylate	Elimination of drug matrix interference, but it is a complicated operation, and requires a special device.	40 ppm	40 ppm	[23]
	IMH ^d^	Simple and avoids drug matrix interference.	1.2 ppm	2.4 ppm	/
TPO ^b^	IMH	Derivatization reagent interference.	2.4 ppm	*	/
BDC ^c^	IMH	Simple, high sensitivity, and avoids drug matrix interference.	0.3 ppm	0.6 ppm	/

^a^ DDTC: sodium *N*,*N*-diethyldithiocarbamate, ^b^ TPO: sodium thiophenolate, ^c^ BDC: sodium dibenzyldithiocarbamate, ^d^ IMH: 2,4,4′-trimethoxy-5,6,5′,6′-bis(methylenedioxy)-2′-morpholine methylbiphenyl methanesulfonate, * derivatization reagent solution has interference, / The data came from our own research.

**Table 2 molecules-27-01950-t002:** Summary report of the method validation.

Parameter	MMS Derivatives	EMS Derivatives
Linear equation	*y* = 133,809*x* + 5669.9	*y* = 105,889*x* − 526.49
R	0.9998	0.9998
Linearity range (μg·mL^−1^)	0.03–3.00	0.03–3.00
LOQ (ppm)	0.3	0.6
Precision% (*n* = 6)	3.23	1.66
Intermediate precision% (*n* = 12)	3.50	2.39
Stability% (24 h)	2.55	2.40
Accuracy at LOQ (*n* = 3)		
recovery%	100.95	100.17
RSD%	4.53	1.79
Accuracy at 80% level (*n* = 3)		
recovery%	100.36	99.15
RSD%	1.88	2.77
Accuracy at 100% level (*n* = 3)		
recovery%	99.2	99.86
RSD%	1.83	1.54
Accuracy at 120% level (*n* = 3)		
recovery%	100.03	99.73
RSD%	4.92	3.66

**Table 3 molecules-27-01950-t003:** Determination results by the two analytical methods.

Batch No.	PGIs	Derivatization HPLC–UV Method	Derivatization GC–MS Method [19]
		ppm	ppm
20180608	MMS	4.56	5.16
	EMS	/	/
20180918	MMS	4.84	5.30
	EMS	/	/
20181026	MMS	5.74	5.84
	EMS	/	/

## Data Availability

The data presented in this study are available on request from the corresponding author.

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
