# Peer review of "Determination of Methyl Methanesulfonate and Ethyl Methylsulfonate in New Drug for the Treatment of Fatty Liver Using Derivatization Followed by High-Performance Liquid Chromatography with Ultraviolet Detection"

_molecules, 2022, doi:10.3390/molecules27061950_

Round 1
Reviewer 1 Report
The manuscript describes determination of methyl methanesulfonate and ethyl methylsulfonate in new drug by HPLC. This protocol uncludes chemical modification on analyte prior the analysis. My overall impression of the work is positive. The study was carried out methodogically correct. Unfortunately, the presented work is, in my opinion, missing the novelty. Recently, "Determination of methyl methanesulfonate and ethylmethanesulfonate in methanesulfonic acid by derivatization followed by high-performance liquid chromatography with ultraviolet detection" has been published (DOI: 10.1002/jssc.201700543) and deals with a very similar issue. The general difference is in a different derivatizing reagent and focusing on the specific drug. Out of this, the novelty of the article is not sufficient for publication in the journal with IF greater than 4. Thus, I do not recommend to publish the manuscript in Molecules.
Below are some suggestions for possible improvements :
- The authors described the the quantitaion of methanosulfonate in drug IMH. This drug will be tested clinically. The Authors wrote: "The bulk of API of IMH (batch nos. 20180608, 20180918, and 20181026) was produced 206 in the authors’ laboratory". So, if the structure is known, it should be placed in manuscript. It allows for better understanding of sample preparation protocol.
- The authors wrote: The determination results of samples were consistent with those obtained from derivatization gas chromatography–tandem mass spectrometry (GC–MS) analysis. Tandem mass spectrometry means the usage of two the same mass analyser e.g. TOF-TOF. If just an ordinary GCMS with e.g single quadrupole mass analysed was used, then you should not desribed it as tandem mass spectometry
- the x-axis unit description represents an invalid alphabet
Author Response
Dear reviewer,
Our article “Determination of methyl methanesulfonate and ethyl methylsulfonate in new drug for the treatment of fatty liver using derivatization followed by high–performance liquid chromatography with ultraviolet detection” (ID: 1620621) has been revised in accordance with your suggestion, and we also read it carefully to correct the mentioned errors. The contents of the revision and the response to the reviewer’s comments are described in detail in the text.
Question 1: The manuscript describes determination of methyl methanesulfonate and ethyl methylsulfonate in new drug by HPLC. This protocol uncludes chemical modification on analyte prior the analysis. My overall impression of the work is positive. The study was carried out methodogically correct. Unfortunately, the presented work is, in my opinion, missing the novelty. Recently, "Determination of methyl methanesulfonate and ethylmethanesulfonate in methanesulfonic acid by derivatization followed by high-performance liquid chromatography with ultraviolet detection" has been published (DOI: 10.1002/jssc.201700543) and deals with a very similar issue. The general difference is in a different derivatizing reagent and focusing on the specific drug. Out of this, the novelty of the article is not sufficient for publication in the journal with IF greater than 4. Thus, I do not recommend to publish the manuscript in Molecules.
Answer 1: Compared with the methods mentioned in the reported literature, our method adopted a new derivatization reagent with a stronger UV absorption chromophore for the first time, which significantly improved the sensitivity of detection. In addition, we used a new and simple sample pretreatment method, which effectively overcomes matrix interference of the drug. From these evidences mentioned above, the method applied in this paper is different from and is superior to those reported in the literature.
In order to display the advantages of the proposed method clearly, the information about features of our new method and other methods published in literature were described in Table 1 of our revised manuscript.
Question 2: The authors described the the quantitaion of methanosulfonate in drug IMH. This drug will be tested clinically. The Authors wrote: "The bulk of API of IMH (batch nos. 20180608, 20180918, and 20181026) was produced 206 in the authors laboratory". So, if the structure is known, it should be placed in manuscript. It allows for better understanding of sample preparation protocol.
Answer 2: The chemical name and structure of IMH have been added in the manuscript in accordance with the suggestion, for details, see lines 45-46 and Figure 1.
Question 3: The authors wrote: The determination results of samples were consistent with those obtained from derivatization gas chromatography tandem mass spectrometry (GC MS) analysis. Tandem mass spectrometry means the usage of two the same mass analyser e.g. TOF-TOF. If just an ordinary GCMS with e.g single quadrupole mass analysed was used, then you should not desribed it as tandem mass spectrometry.
Answer 3: This is a wrting mistake, in light of the reviewer’s suggestion, the description “gas chromatography-tandem mass spectrometry” has been corrected into “gas chromatography-mass spectrometry” in abstract section, see line 23.
Question 4: the x-axis unit description represents an invalid alphabet
Answer 4: We used a data processing system with a Chinese interface. Thus, in Fig. 4 and Fig. 8, the unit of the X-axis displays Chinese characters “分钟”, which mean “minutes” (a time unit). We have added an English annotation beside the Chinese characters. See Fig. 4 and Fig. 8, in detail.
Yours sincerely,
Qingyun Yang
Mar 9, 2022

Reviewer 2 Report
My comments and suggestions for authors are given at the attachment.

Author Response
Dear reviewer,
Thanks for your suggestion, we hope this paper is of interest for readers.
Our article “Determination of methyl methanesulfonate and ethyl methylsulfonate in new drug for the treatment of fatty liver using derivatization followed by high–performance liquid chromatography with ultraviolet detection” (ID: 1620621) has been revised in accordance with your suggestion, and we also read it carefully to correct the mentioned errors. The contents of the revision and the response to the reviewer’s comments are described in detail in the text.
Question 1: L. 45. What acronym IMH means?
Answer 1: “IMH” is the code name for the innovative drug, which is the test sample in this study The chemical name and chemical structure of IMH have been added in the revised manuscript, for details, see lines 45-46 and Fig. 1.
Question 2: L. 133. Why the derivatization time is 2 h? According to fig. 5A 1 h is quite enough.
Answer 2: Considering that the derivatization time range investigated was between 1 h~3 h, and the derivatization duration time less than 1 h was not investigated. To ensure the full progress of the derivatization reaction, we used 2 h as the reaction time. For future practical applications, we will consider shortening the reaction time according to the actual circumstances to improve efficiency.
Question 3: Fig. 4. What are the not described peaks?
Answer 3: In Fig.4, the not described peaks are signals of derivatization reagent (BDC), and the by-product of the derivatization reaction. They have been marked in the revised manuscript, see Fig. 4 in detail.
Question 4: Fig. 7. A. Sample-spiked with what?
Answer 4: “Sample-spiked” is a wrting mistake, it should be “spiked sample”. It means “an appropriate amount of standard (MMS and EMS) were added to the sample (IMH) solution”. It has been modified, see line 186 and 206 in detail.
Question 5: Fig. 7 B. The chromatogram is almost exactly the same as A.
Answer 5: In Figure 8 of the revised manuscript (figure 7 in the original manuscript), figure A is the chromatogram of “spiked sample” solution with the signals of the sample (Rt ~ 3-4 minutes), while figure B is the chromatogram of “impurity” standard solution without the signal of the sample. Since the concentrations of EMS and MMS in both solutions were almost identical, the signals of the two derivatized products were almost exactly the same.
Question 6: Table 2. How was LOQ determined and what is its value?
Answer 6: LOQ values were determined as follows: “Precisely measure the appropriate amount of 30·ng mL–1 solution and dilute with ACN stepwise. LOQs were defined as the concentrations that could be detected and yield signal-to-noise (S/N) ratios of 10 :1.” LOQ values of MMS and EMS derivatives were 0.15 ng mL–1 and 0.30 ng mL–1, equivalent to 0.3 ppm and 0.6 ppm, respectively. We have added the corresponding description in the revised manuscript, see lines 286-288, and lines 193-195, and Table2.
Question 7: What was the method of quantitative analysis used in this work?
Answer 7: In this study, the contents of MMS and EMS in the test solution were calculated by the reference method (see line 264-265). That is, for each impurity, the concentration of MMS and EMS in standard solution was used to calculate the percentage contents in test solution.
Question 8: L. 199. What PGI acronym means?
Answer 8: PGI means potential genotoxic impurity. In this study, EMS and MMS are potential genotoxic impurities (PGIs). All of the PGIs in revised manuscript have been checked.
Question 9: Point 3.3.2. The description is rather unclear. It should be clearly said what what kinds of separate mixtures were prepared.
Answer 9: In view of the reviewer’s suggestion, we have optimized the description about derivatization procedure in section 3.3.2. For details, see lines 256-265.
Question 10: L. 295 „…the two methods were almost identical.” and L. 297 „…a versatile and convenient method…”. The HPLC and GC methods should be compared in detail. It is known that GC is cheap and quick method of analysis. What are the advantages and disadvantages of the two methods in the described analysis?
Answer 10: To examine methanesulfonate genotoxic impurities, the European Pharmacopoeia adopts the derivatization-GC-MS method, and the derivatization-HPLC-UV is proposed in this paper. Both methods require pre-column derivatization, and the determination results were almost identical. Compared with mass spectrometry detectors, UV detectors are less expensive and more popular. The derivatization-HPLC-UV method proposed in this paper can be a convincing complement to the European Pharmacopoeia method. Researchers can alternately use these two methods on a case-by-case basis. The corresponding description of “this method could be used as a convincing supplement to GC-MS method for the determination of methanesulfonate genotoxic impurities in pharmaceuticals” has been added. The details are shown on line 306-308.
Yours sincerely,
Qingyun Yang
Mar 9, 2022

Reviewer 3 Report
The article is very well written with very precise objective. The article is very well organized. besides, figures and tables are very clear and well represented, thanks. However, many revisions are required
- IMH full name should be mentioned in the first time mentioning line 45
- Line 50 , remove extra g ; typo error
- Chemical structures for the main investigated compounds should be represented as fig 1. And it should be referred in line 69.
- Full names for dervatization agents should be stated e.g. DDTC, TPO, and BDC.
- Full names for aprotic solvents, including ACN, DMF, DMA, and DMSO, should be stated too. Line 113
- Linearity range in table 2, should be written as 0.03-3.00 ( number of significant figures should be the same ) . correct it also in line 260
- Why did the authors select 280 nm for measurement ??? it is strongly recommended to provide a figure for UV spectra for estimated pure impurities alone, derivatizing agent, and MMS derivative / EMS derivative as superimposed spectra in one figure. Or at least the spectrum for MMS/EMS derivatives.
- Add reference for GC/MS method in Table 3. Also for lines 227 and 276, GC/MS analysis.
- What is the meaning of Wahaha in line 217?
- Abstract and conclusion should be written without abbreviations
- Conclusion is too long. Go for the main novel findings only. I suggest removing the following sentences [ lines 289 till 292] “Various key experimental parameters, such as deri-289 vatization solvent, reaction temperature, reaction time, and concentration of derivatiza-290 tion solution, were also explored and optimized. The accuracy and reliability of the pro-291 posed method were confirmed through comprehensive method verification in accordance 292 with ICH guidelines.”
- Add study limitation and future research at the end of the discussion..e.g.
Testing other pharmaceuticals
Identification of unknown peak at RT 12.5 using MS
Suggest solution or rebuttal for reduction of long dervatization reaction time ( 2 hours )
……..the authors can add more comments here.
Best wishes
Author Response
Dear reviewer,
Thanks for your suggestion, we hope this paper is of interest for readers.
Our article “Determination of methyl methanesulfonate and ethyl methylsulfonate in new drug for the treatment of fatty liver using derivatization followed by high–performance liquid chromatography with ultraviolet detection” (ID: 1620621) has been revised in accordance with your suggestion, and we also read it carefully to correct the mentioned errors. The contents of the revision and the response to the reviewer’s comments are described in detail in the text.
Question 1: IMH full name should be mentioned in the first time mentioning line 45
Answer 1: In light of the reviewer’s suggestion, the chemical name and structure of IMH have been added in the revised manuscript, for details, see lines 45-46 and Figure. 1.
Question 2: Line 50 , remove extra g ; typo error
Answer 2: The word “druggability” is an academic term, it is used in drug discovery to describe a biological target that is known to or is predicted to bind with high affinity to a drug. It is a right word, not typo error.
Question 3: Chemical structures for the main investigated compounds should be represented as fig 1. And it should be referred in line 69.
Answer 3: The chemical structures and names of IMH, MMS, and EMS have been represented in the revised manuscript (see Fig. 1 in details) according to the suggestion.
Question 4: Full names for dervatization agents should be stated e.g. DDTC, TPO, and BDC.
Answer 4: In accordance with the suggestion, the full names and chemical structures for dervatization agents have been represented in Fig. 1 and lines 97-98.
Question 5: Full names for aprotic solvents, including ACN, DMF, DMA, and DMSO, should be stated too. Line 113
Answer 5: In light of the reviewer’s suggestion, the full names for aprotic solvents have been stated in the revised manuscript, see lines 118-119 in detail.
Question 6: Linearity range in table 2, should be written as 0.03-3.00 ( number of significant figures should be the same ) . correct it also in line 260
Answer 6: The value of linearity rang has been corrected into “0.03-3.00” in section 3.3.3 (line 272) and table 2 in the light of the reviewer’s suggestion.
Question 7: Why did the authors select 280 nm for measurement ??? it is strongly recommended to provide a figure for UV spectra for estimated pure impurities alone, derivatizing agent, and MMS derivative / EMS derivative as superimposed spectra in one figure. Or at least the spectrum for MMS/EMS derivatives.
Answer 7: The maximum absorption wavelengths of MMS and EMS derivatives were 279.3 nm and 281.7 nm (Figure 7). Thus, 280 nm was selected as the determination wavelength. According to the reviewer’s opinion, the UV spectra for MMS/EMS derivatives have been added as Figure.7, and the corresponding description have been added in section 2.2.1 (lines 179-181).
Question 8: Add reference for GC/MS method in Table 3. Also for lines 227 and 276, GC/MS analysis.
Answer 8: The appropriate reference (19) for GC-MS method has been added into the text in accordance with the suggestion, for details, see Table 3, line 239, line 290, and references [19].
Question 9: What is the meaning of Wahaha in line 217?
Answer 9: Wahaha is a brand of purified water in China. The corresponding description have been modified in section 3.1 (lines 228-229).
Question 10: Abstract and conclusion should be written without abbreviations
Answer 10: According to the reviewer’s opinion, we have checked and corrected the abbreviations in abstract carefully. However, we think the abbreviations of the discussion should be kept, otherwise this section seems too long.
Question 11: Conclusion is too long. Go for the main novel findings only. I suggest removing the following sentences [ lines 289 till 292] “Various key experimental parameters, such as deri-289 vatization solvent, reaction temperature, reaction time, and concentration of derivatiza-290 tion solution, were also explored and optimized. The accuracy and reliability of the pro-291 posed method were confirmed through comprehensive method verification in accordance 292 with ICH guidelines.”
Answer 11: According to the reviewer’s suggestion, we have removed these sentences, and optimized the conclusion in the manuscript.
Question 12: Add study limitation and future research at the end of the discussion..e.g. Testing other pharmaceuticals
Answer 12: The method proposed herein can be used for the determination of genotoxic impurities of other drugs containing methanesulfonate, but no specific experimental data are present on the corresponding applications. The corresponding description in the paper as follows: “As a versatile and convenient method, the proposed method has high reference value for the quality control of other mesylate drugs”. The details are presented on line 309-311.
Question 13: Identification of unknown peak at RT 12.5 using MS
Answer 13: The unkown peaks at RT 12.5 min were the signal of the by-product of the derivatization reaction. Because the sample contains a strong base NaOH, there is a risk of contamination of the mass spectrometer detector, and the MS method is not used to identify its structure.
Question 14: Suggest solution or rebuttal for reduction of long dervatization reaction time ( 2 hours )
Answer 14: Considering that the derivatization time range investigated was between 1 h ~ 3 h, and the derivatization duration time less than 1 h was not investigated. To ensure the full progress of the derivatization reaction, we used 2 h as the reaction time. For future practical applications, we will consider shortening the reaction time according to the actual circumstances to improve efficiency.
Yours sincerely,
Qingyun Yang
Mar 9, 2022

Round 2
Reviewer 1 Report
Dear Authors,
Thank you for your answers and for considering my suggestions. Taking into account your justification, I accept it and recommend this manuscript for publication.
Author Response
Dear reviewer,
Thanks for your suggestion, we hope this paper is of interest for readers.
Yours sincerely,
Qingyun Yang
Mar 13, 2022

Reviewer 3 Report
THANKS for all your work and responses . The paper is ready for publication now.
Greetings
Author Response

(The authors gave the same response as above.)
